

# Direct measurement of $N_2O_5$ heterogeneous uptake coefficients on atmospheric aerosols in southwestern China and evaluation of current parameterizations

Jiayin Li [a⊥], Tianyu Zhai [a,b⊥], Xiaorui Chen [c*], Haichao Wang [c], Shuyang Xie [a], Shiyi Chen [a], Chunmeng Li [a,d], Huabin, Dong [a], Keding Lu [a*]

[a] State Key Joint Laboratory of Environmental Simulation and Pollution Control, College of Environmental Science and Engineering, Peking University, Beijing 100871, China

[b] State Environmental Protection Key Laboratory of Vehicle Emission Control and Simulation, Chinese Research Academy of Environmental Sciences, Beijing, 100012, China

[c] School of Atmospheric Sciences, Sun Yat-sen University, and Southern Marine Science and Engineering Guangdong Laboratory (Zhuhai), Zhuhai, 519082, China

[d] The National Institute of Metrology, Center for Environmental Metrology, Beijing 100029, China

[⊥]Jiayin Li and Tianyu Zhai contributed equally in this work.

**Corresponding author:** Xiaorui Chen (chenxr95@mail.sysu.edu.cn) and Keding Lu (k.lu@pku.edu.cn)

**Abstract:** The heterogeneous hydrolysis of dinitrogen pentoxide ($N_2O_5$) is a critical process in assessing $NO_x$ fate and secondary pollutants (e.g. particulate nitrate) formation. However, accurate quantification of the $N_2O_5$ uptake coefficient ($\gamma(N_2O_5)$) in the ambient conditions is a challenging problem which can causes unpredictable uncertainties in the predictions of the air quality models. Here, the $\gamma(N_2O_5)$ values were directly measured using an improved aerosol flow tube system in a city located on the plateau in southwestern China to investigate its influencing factors and the performance of current $\gamma(N_2O_5)$ parameterization under this typical environmental condition. The nocturnal mean $\gamma(N_2O_5)$ value ranged from 0.0018 to 0.12 with an average of 0.023±0.021. The aerosol water significantly promoted $N_2O_5$ uptake, while particulate organic and nitrate generally showed suppression effect. We found that median $\gamma(N_2O_5)$





predicted by some parameterizations agreed well with observation, whereas the
parameterizations failed to reproduce the range of observed values and showed poor
correlations ($R^2$=0.00~0.09). Elevated differences between prediction and observation
specifically occurred at high aerosol liquid water content (ALWC) with an
underestimation by -37%~-1% and low ALWC with an overestimation by 34~189%,
respectively. Such differences between the measured and parameterized $\gamma(N_2O_5)$ would
lead to biased estimation (-77%~74%) on particulate nitrate production potential. Our
findings suggest the need for more direct field quantifications of $\gamma(N_2O_5)$ and the
laboratory measurements under extreme ALWC conditions to re-evaluate the response
coefficients between $\gamma(N_2O_5)$ and aerosol chemical compositions.
**Keywords:** $N_2O_5$ uptake coefficient, $\gamma(N_2O_5)$ parameterizations, particulate nitrate
formation, nighttime chemistry

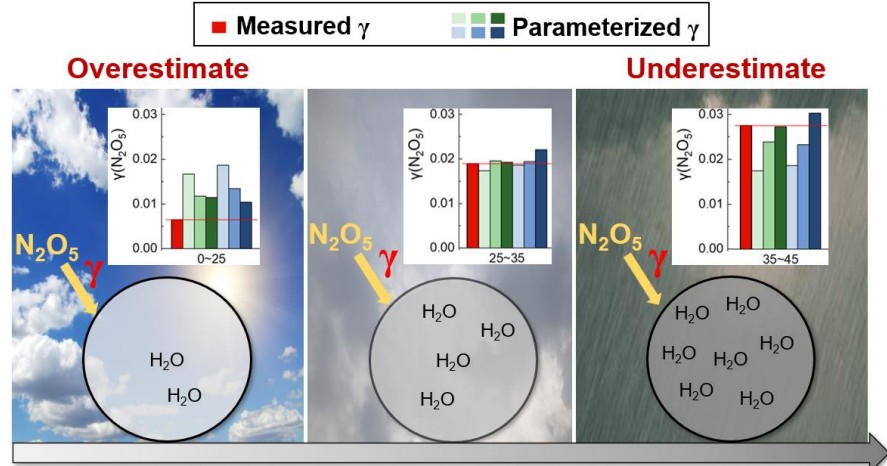


**Graphic Abstract**


## 1. Introduction

Nitrate radical ($NO_3$) and dinitrogen pentoxide ($N_2O_5$) are dominant in nocturnal
atmospheric chemistry as reactive nitrogen species that can strongly influence the
concentration and distribution of ozone ($O_3$) and nitrogen oxides ($NO_x$=$NO$+$NO_2$), and




the air quality (Brown et al., 2006;Wang et al., 2023;Decker et al., 2019;Dentener and
Crutzen, 1993). $NO_3$ is produced by the reaction of $NO_2$ and $O_3$ (R1), and there is a
thermodynamic equilibrium between $NO_3$ and $N_2O_5$ (R2), which is the source of $N_2O_5$
(Brown and Stutz, 2012). There are two main pathways for $NO_3$ removal: the direct one
is reactions of $NO_3$ and VOCs (R3), especially alkenes, and the indirect way is the
heterogenous hydrolysis of $N_2O_5$ (Asaf et al., 2009;Ng et al., 2017). $N_2O_5$ can react
with $H_2O$ and chloride ($Cl^-$) in the particle phase and form soluble nitrate and nitryl
chloride ($ClNO_2$) (R4) (Osthoff et al., 2008;Chang et al., 2011). The uptake of $N_2O_5$ is
the main pathway for the formation of particulate nitrate at night, which contributes to
$PM_{2.5}$ (<2.5 μm in diameter) pollution. Meanwhile, chlorine radical is produced by
$ClNO_2$ photodecomposition in daytime and further regulate the $O_3$ pollution production
by promoting the oxidation of VOCs (Finlaysonpitts et al., 1989;Riedel et al., 2014).
Thus, it is important to quantify the rate of the $N_2O_5$ heterogeneous hydrolysis reaction
in ambient conditions.

$$NO_2 + O_3 \ \rightarrow \ NO_3 + O_2 \qquad\qquad (R1)$$
$$NO_3 + NO_2 + M \leftrightarrow N_2O_5 + M \qquad\qquad (R2)$$
$$NO_3 + VOCs \ \rightarrow \ Products \qquad\qquad (R3)$$
$$N_2O_5 + H_2O/Cl^-(p) \ \rightarrow \ NO_3^-(p) + ClNO_2 \qquad\qquad (R4)$$

$γ(N_2O_5)$ is defined as the net probability of $N_2O_5$ irreversibly taken up onto an
aerosol surface upon collision (McDuffie et al., 2018). According to previous study, the
process of $N_2O_5$ heterogeneous hydrolysis reaction on aerosols was treated as a resistor
model including three steps: gas diffusion (R5), surface accommodation, and aqueous
reaction (R6~R8) (Abbatt et al., 2012;Fang et al., 2024). This process can be influenced
by aerosol chemical compositions (e.g. aerosol liquid water content (ALWC), nitrate
($NO_3^-$) concentration, $Cl^-$ concentration, and organics), morphology and ambient
meteorological factors (Bertram and Thornton, 2009;Mozurkewich and Calvert,
1988;Roberts et al., 2009;Thornton et al., 2003). High concentration of ALWC and $Cl^-$
can promote the uptake reaction (R6~R8), and $NO_3^-$ suppress the reaction (R6).
Organics also can suppress the reaction by forming a coating on the surface of the
particles and regulating the ALWC and the passage rate of $N_2O_5$ molecules (Folkers et
al., 2003;Gaston et al., 2014;Anttila et al., 2006). However, the above results are mainly
based on laboratory studies. In ambient conditions, the correlations between $γ(N_2O_5)$
and aerosol chemical compositions were generally weak mainly due to the coupling





effects of particle morphology, size, mixing state, and meteorological parameters (e.g.
temperature and relative humidity) (Phillips et al., 2016;Wang et al., 2020b;Riedel et
al., 2012).

$$N_2O_5 \text{ (g)} \leftrightarrow N_2O_5 \text{ (aq)} \tag{R5}$$

$$N_2O_5(aq) + H_2O(l) \leftrightarrow H_2ONO_2^+(aq) + NO_3^- \text{ (aq)} \tag{R6}$$

$$H_2ONO_2^+ \text{ (aq)} + H_2O(l) \rightarrow HNO_3(aq) + H_3O^+ \text{ (aq)} \tag{R7}$$

$$H_2ONO_2^+ \text{ (aq)} + HX(aq) \rightarrow XNO_2(aq) + H_3O^+ \text{ (aq)} \tag{R8}$$

In order to accurately quantify the contribution of $N_2O_5$ heterogeneous hydrolysis

to nitrate formation and $NO_x$ regulation, a variety of parameterizations of $\gamma(N_2O_5)$ have
been established based on laboratory and field studies (Evans and Jacob, 2005;Davis et
al., 2008;Yu et al., 2020;Bertram and Thornton, 2009). The parameters in
parameterizations mainly include the meteorological parameters, concentrations of
aerosol chemical compositions, and particle physicochemical parameters. However, the
comparisons of parameterized and measured $\gamma(N_2O_5)$ in field measurements revealed
significant discrepancies between them (Brown et al., 2009;Ryder et al.,
2014;McDuffie et al., 2018), which mainly lie in the large variations in response of
$\gamma(N_2O_5)$ to particle compositions on ambient particles. Moreover, the overestimation or
underestimation of the parameterized $\gamma(N_2O_5)$ can leads to unpredictable biases in the
simulations of the chemical transport models (Murray et al., 2021;Chen et al.,
2018;Ryder et al., 2014).

Until now, only a few studies have quantified $\gamma(N_2O_5)$ values in ambient conditions

($<10^{-4}$ to 0.1) mostly by indirect quantification methods (Brown et al., 2016;Wang et
al., 2018;Chen et al., 2020b;Morgan et al., 2015;Tham et al., 2018) while some by direct
measurements (Yu et al., 2020;Riedel et al., 2012;Bertram et al., 2009a). The $N_2O_5$
heterogeneous uptake process has been reported to be active in China. The $\gamma(N_2O_5)$
values in North China Plain, Yangtze River Delta and Pearl River Delta in China ($10^{-2}\sim10^{-1}$)
were generally about 1 to 2 orders of magnitude larger than that in European
and North America ($10^{-3}\sim10^{-2}$) (Yan et al., 2023;Wang et al., 2017b;Wang et al.,
2017d;Wang et al., 2017c;Niu et al., 2022). To further investigate the $N_2O_5$
heterogeneous chemistry in China, the $\gamma(N_2O_5)$ values were directly measured in a
typical highland city, Kunming, in China using an improved aerosol flow tube system
from 15 April to 20 May 2021. The relationship between the $\gamma(N_2O_5)$ values and
impacting factors was determined. We then examine the performance of current $\gamma(N_2O_5)$



parameterizations by comparing to the observed values and analyze the causes of
discrepancies in extreme ALWC conditions. We further notice the significant biases of
particulate nitrate formation potential estimated by $\gamma(N_2O_5)$ parameterization.
**2. Methods**
**2.1. Site description**

The field campaign was conducted in Kunming, China from 15 April to 20 May

2021. The main sampling site was on the roof of the Yihe Building, Yijingyuan Hotel
(24°59′05″ N, 102°39′40″ E), about 20 m above the ground. As shown in Figure 1, the
measurement site was located approximately 1890 m above sea level, 8 km away from
the city center, and 1 km from Dianchi Lake to the west. The site receives traffic
emissions from two roads within a radius of 500 m. The site was mainly surrounded by
residential area and there was no major industrial source around. Besides, the particle
composition was measured at Guandu Forest Park (25°00′43″ N, 102°45′55″ E), which
was about 9 km away from Yijingyuan Hotel, 5.2 km from the city center and was also
mainly surrounded by residential living area. Sunrise was around 06:30 CNST and
sunset at 19:30 CNST.

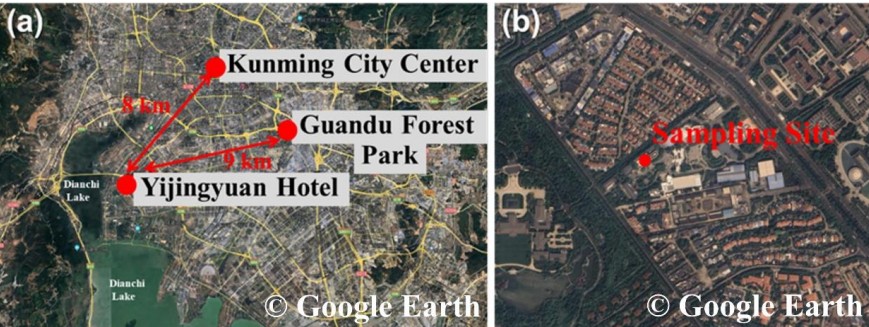


**Figure 1.   Google Maps images showing the locations of the experimental sites.**
(a) The location of the Yijingyuan Hotel and Guandu Forest Park. (b) The surrounding
environment of Yijingyuan Hotel.
**2.2. Instrument setup**

Multiple gas phase and particulate parameters were measured during the campaign,

including $N_2O_5$, NO, $NO_2$, $O_3$, VOCs, $PM_{2.5}$, particle number size distribution (PNSD),
particle composition, and meteorological parameters. The detailed information of the
instruments is listed in Table 1.



N$_2$O$_5$ concentration was measured by a cavity-enhanced absorption spectrometer
(CEAS) developed by Wang et al. (Wang et al., 2017a), and has been used in several
field campaigns. N$_2$O$_5$ in the sampling gas was thermally decomposed to NO$_3$ in a
preheated perfluoroalkoxy alkane (PFA) tube (130 $^o$C), and then detected in a resonator
cavity maintained at 110 $^o$C to avoid the reversible reaction of N$_2$O$_5$ and NO$_3$. Excess
NO was injected to the cavity every 5 min to obtain the reference spectrum by
eliminating the influence of water vapor. The N$_2$O$_5$ loss in the sampling system and
detection system were also calibrated and corrected during data processing. The
detection of limit (LOD) of CEAS was 2.7 pptv (1$\sigma$), and the uncertainty was 19%.
NO, NO$_2$ and O$_3$ were monitored by commercial instruments (Thermo-Fisher 42i
and 49i). A total of 117 kinds of volatile organic compounds (VOCs) were measured by
an automated gas chromatograph equipped with a mass spectrometer and flame
ionization detecter (GC-MS/FID). The particle composition was measured by a time-
of-flight aerosol chemical speciation monitor (ToF-ACSM), including sulfate, nitrate,
ammonium, chloride and organics. The ALWC was calculated by ISORROPA-II model
and did not consider the hygroscopicity of organic compounds (Fountoukis and Nenes,
2007). PNSD were measured by a scanning mobility particle sizer (SMPS, TSI Model
3938) including an Electrostatic Classifier (Model 3082) and a condensation particle
counter (CPC, Model 3776). Meteorological parameters, included relative humidity
(RH), temperature (T), pressure, wind speed and wind direction, were available during
the campaign.

Table 1. The detailed information of instruments during the campaign.

| Parameters | Detection of limit | Method | Accuracy |
|---|---|---|---|
| N$_2$O$_5$ | 2.7 pptv (1$\sigma$, 1min) | CEAS | $\pm$ 19% |
| NO | 50 pptv (2min) | Chemiluminescence | $\pm$ 10% |
| NO$_2$ | 50 pptv (2min) | Chemiluminescence[a] | $\pm$ 10% |
| O$_3$ | 0.5 ppbv (2$\sigma$, 1min) | UV photometry | $\pm$ 5% |
| VOCs | 2–190 ppt (1 h) | GC-MS/FID | $\pm$ 5% |
| PNSD | 14–730 nm (5 min) | SMPS | $\pm$ 10% |
| Particle composition | m/z 10 – 219 (10 min) | ToF-ACSM | - |
| $\gamma$(N$_2$O$_5$) | 0.0016 (40 min) | Aerosol flow tube system | $\pm$ 16~43 % |

[a] Photolytic conversion to NO through blue light before detection.





**2.3. The measurement and calculation of γ($N_2O_5$)**

The γ($N_2O_5$) was directly measured by an aerosol flow tube system (AFTS) coupled with a detailed box model developed by Chen et al. (Chen et al., 2022). The detection limit and accuracy of the AFTS are listed in Table 1. Briefly, the AFTS mainly consists of a $N_2O_5$ generator, an aerosol flow tube, and detection instruments for $N_2O_5$, $NO_x$, $O_3$ and $S_a$. $N_2O_5$ generated by $O_3$ and $NO_2$ (excess) was added to the sampling gas in the front of the aerosol flow tube. The aerosol flow tube consist of two cones at both ends with a vertex angle of 15° and a straight cylinder in the middle with an inner diameter of 140 mm and a length of 343 mm. The total flow rate in the tube was 2.08 L min$^{-1}$, and the residence time was 259 s. The detection instruments used in this study were Thermo 42i-TL to detect NO and $NO_2$ concentration, Teledyne T265 to detect $O_3$ concentration, CEAS-PKU to detect $N_2O_5$ concentration and SMPS (TSI Model 3938) to detect aerosol surface concentrations ($S_a$). Meanwhile, a RH&T sensor (Rotronic, Model HC2A-S) was used to detect relative humidity and temperature in the flow tube. In a duty cycle, the $N_2O_5$ concentrations with or without aerosols were acquired at both the inlet and exit of the flow tube, NO, $NO_2$ and $O_3$ concentrations were always acquired at the inlet, $S_a$ concentration always acquired at the exit. The loss rate coefficients of $N_2O_5$ were calculated by a time-dependent box model coupled with $NO_3$-$N_2O_5$ chemistry under the constraint of the measurement of $N_2O_5$ concentrations and other auxiliary parameters to overcome the influence of homogeneous reactions (e.g., $NO_2$, $O_3$ and NO) and variations of air mass on γ($N_2O_5$) retrieval. The $N_2O_5$ loss rate in the absence of aerosols was expected as wall loss rate coefficients ($k_{het}^{wo/aerosols}$) of $N_2O_5$, and the loss rate in the presence of aerosols was expected as the loss rate both on wall and aerosols ($k_{het}^{w/aerosols}$) of $N_2O_5$. Therefore, γ($N_2O_5$) could be calculated by Eq (1). Among them, the loss of $S_a$ concentration in aerosol flow tube was corrected by the penetration efficiency derived in our previous study (Chen et al., 2022) and the dry-state $S_a$ were corrected to ambient (wet) $S_a$ by a hygroscopic growth factor (Liu et al., 2013). A stringent data QA/QC procedure is applied prior to model calculation based on above measured variables to retrieve robust γ($N_2O_5$) values. Other detailed information about this system can be found in Chen et al. 2022.





$$\gamma(N_2O_5) = \frac{4 \times (k_{het}^{w/aerosols} - k_{het}^{wo/aerosols})}{c \times S_a} \tag{1}$$

**2.4 The calculation of NO₃ and N₂O₅ reactivity**


NO$_3$ production rate (P(NO$_3$)) was calculated by measured NO$_2$ concentration and
O$_3$ concentration via Eq.(2), $k_{NO_2+O_3}$ represents the reaction rate of constant of NO$_2$
and O$_3$ (Atkinson et al., 2004). NO$_3$ concentration can be calculated by measured N$_2$O$_5$
concentration with the temperature-dependent equilibrium relationship (Eq.3). The
steady-state lifetime of N$_2$O$_5$ ($\tau$(N$_2$O$_5$)) and NO$_3$ ($\tau$(NO$_3$)) was calculated by
concentrations and P(NO$_3$) as shown in Eq.(4) and Eq.(5) (Brown and Stutz, 2012). The
NO$_3$ reactivity with VOCs ($k$(NO$_3$)) can be calculated by Eq.(6), among them $k_i$
represents the bimolecular rate coefficients.

$$P(NO_3) = k_{NO_2+O_3}[NO_2][O_3] \tag{2}$$

$$[NO_3] = [N_2O_5]/k_{eq}[NO_2],$$
$$k_{eq} = 5.5 \times 10^{-27} \times e^{10724/T} \tag{3}$$

$$\tau(N_2O_5) = [N_2O_5]/P(NO_3) \tag{4}$$

$$\tau(NO_3) = [NO_3]/P(NO_3) \tag{5}$$

$$k(NO_3) = \sum k_i[VOC_i] \tag{6}$$

**2.5 The calculation of nitrate production rate**


The N$_2$O$_5$ uptake for nighttime particulate nitrate production is regarded as a
pseudo first order reaction, the rate constant (k$_{N2O5}$) of which can be calculated from
Eq 7 with measured or parameterized $\gamma$(N$_2$O$_5$), where C is the mean molecular speed of
N$_2$O$_5$. The yield ratio of ClNO$_2$ ($\varphi$) was set as a constant of 0.5 in all calculations, which
is consistent with the previously observed yield range 0.3~0.73 in North China (Wang
et al., 2017d;Wang et al., 2018). The nitrate production rate can be calculated by Eq 8,
where [N$_2$O$_5$] is the concentration of N$_2$O$_5$.

$$k_{N2O5}=0.25 \times S_a \times \gamma(N_2O_5) \times C \tag{7}$$

$$P(NO_3^-)= k_{N2O5} \times [N_2O_5] \times (2-\varphi) \tag{8}$$



## 3. Results and discussion

### 3.1. $\gamma(N_2O_5)$ measurement overview and comparison

The mean diurnal of measured $N_2O_5$ concentration, $\gamma(N_2O_5)$ values, RH, T, concentrations of $NO_2$, $O_3$, NO, $PM_{2.5}$ from 15 April to 20 May 2021 are shown in Figure 2a, and the time series are shown in Figure S1. Higher $PM_{2.5}$ concentration was observed at night (average of $27.8\pm14.3$ ug/m$^3$, peak of 81.0 ug/m$^3$) than that in the day (Figure 2a & Figure S1). The $NO_2$ (average of 6.5±8.4 ppbv) and $O_3$ (average of $45.5\pm19.7$ ppbv) concentration in Kunming are lower than other regions in China (Wang et al., 2017d;Wang et al., 2020a;Niu et al., 2022;Li et al., 2020), indicating a lower atmospheric oxidation capacity. The mean nocturnal $NO_3$ production rate ($PNO_3$) was 0.6± 0.8 ppbv/h, which is also lower than previous reports in China (Tham et al., 2016;Zhai et al., 2023;Wang et al., 2022). During this observation campaign, significant $N_2O_5$ concentration (at a maximum of 395.1 pptv) was only observed within April 16-27 mainly with low humidity and high precursor concentrations, while the concentrations fluctuated around the detection limit during other periods. The nocturnal mean concentration of $N_2O_5$ was $33.4\pm75.2$ pptv, which is lower than reported concentrations in other regions of China (Wang et al., 2018;Brown et al., 2016;Zhai et al., 2023). During the field measurement, high temperature (~20℃) favors the equilibrium shifting from $N_2O_5$ towards $NO_3$ and site mainly received the emissions from vegetations in the surrounding parks. In that case, the major removal of $NO_3$-$N_2O_5$ at night was the reaction of $NO_3$ with VOCs represented by monoterpene (67%) and isoprene (4%), followed by $N_2O_5$ uptake (15%) shown in Figure 2c. Rapid depletion of daytime emitted isoprene by $NO_3$ led to low contribution of isoprene to $NO_3$ reactivity after sunset (Figure S2). The steady-state lifetime of $N_2O_5$ ($\tau(N_2O_5)$) was $185\pm294$ s on average and its diel pattern was similar to $N_2O_5$ concentration. The $\tau(N_2O_5)$ in Kunming were higher than most other cities in China (Wang et al., 2020a;Li et al., 2020;Yan et al., 2019). Comparisons of $NO_3$ and $N_2O_5$ concentrations, P($NO_3$), and other parameters with that recently reported in other regions across the world are summarized in Table S1.

The nocturnal mean $\gamma(N_2O_5)$ value ranged from 0.0018 to 0.12 with an average of 0.023±0.021. The diurnal profiles showed that the $\gamma(N_2O_5)$ value decreased after sunset and then sharply increased with relative humidity after midnight, peaking at 5:00 am (Figure 2a). The mean $\gamma(N_2O_5)$ was lower than that in North China Plain and Eastern





China, and similar to that in Pearl River Delta China, Europe and North America
(Figure.2b) (Yan et al., 2023;Wang et al., 2017b;Wang et al., 2017d;Wang et al.,
2017c;Niu et al., 2022;Morgan et al., 2015;Phillips et al., 2016;Bertram et al.,
2009a;McDuffie et al., 2018). The detailed comparisons of field derived γ(N₂O₅) were
summarized in Table S2.

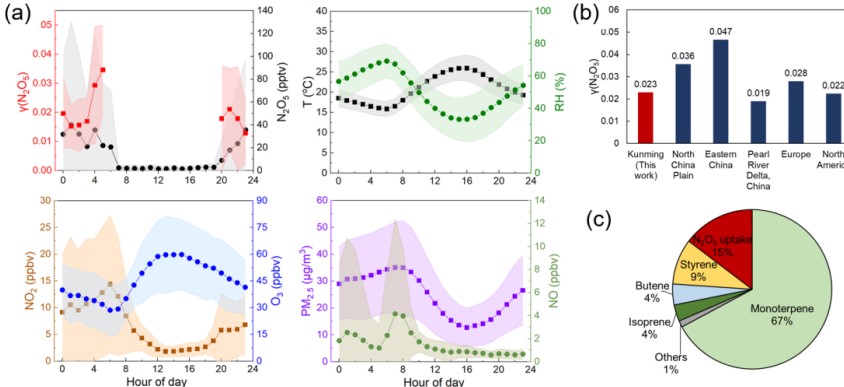


**Figure 2. The overview of γ(N₂O₅), gas phase and particulate parameters,**
**meteorological parameters and NO₃ loss pathways.** (a) Mean diurnal profiles of
measured γ(N₂O₅), N₂O₅, T, RH, NO₂, O₃, PM₂.₅ and NO. (b) Comparison of γ(N₂O₅)
values in China, Europe, and North America calculated from previous work with
measured value in this work. (c) The percentage of NO₃ loss pathway via VOCs and
N₂O₅ uptake at night.
**3.2. Functional dependence of measured γ(N₂O₅) values**

The dependence of measured γ(N₂O₅) values on organics, ALWC, NO₃⁻ and Cl⁻

concentration in particle phase in this study are shown in Figure 3. The organic wet
mass fraction showed a significant negative correlation (R²=0.83) with measured
γ(N₂O₅) values (Figure 3a), indicating that organics in the aerosol significantly
inhibited the uptake of N₂O₅ during the measuring period in Kunming. While a large
number of studies have observed evident suppression of particulate organic on N₂O₅
uptake on lab-generated aerosols (Escoreia et al., 2010;Cosman and Bertram,
2008;Gaston et al., 2014), the negative correlation of particulate organic and γ(N₂O₅)
was usually weak derived from field measurements (Brown et al., 2009;McDuffie et al.,
2018;Chen et al., 2018;Wang et al., 2020b).

Aerosol liquid water also exhibited controlling role on heterogeneous uptake of

N₂O₅ in this study as demonstrated by the evidently positive correlation (R²=0.74) of



ALWC and $\gamma(N_2O_5)$ (Figure 3b). A weak correlation was observed with ALWC below
25 M and a significant correlation observed with ALWC higher than 25 M. The similar
trend has been reported by previous laboratory studies (Mozurkewich and Calvert,
1988;Bertram and Thornton, 2009;Folkers et al., 2003;Hallquist et al., 2003). When RH
is low, the aerosols mainly exist in solid state with low ALWC, limiting the uptake
reaction. Whereas the aerosols become deliquesced as the RH (also ALWC) increases,
which greatly promote the uptake reaction. Previous field studies also found good
correlations of $\gamma(N_2O_5)$ values with ALWC or RH in most regions in China, indicating
that ALWC may be one of the rate-limiting steps of heterogeneous reaction in China
(McDuffie et al., 2018;Yu et al., 2020;Tham et al., 2018;Wang et al., 2022).
Figure 3c showed the negative dependence of measured $\gamma(N_2O_5)$ values on aerosol
nitrate concentration, similar to the results of previous laboratory studies and most field
observations (Tham et al., 2018;Bertram et al., 2009b;Morgan et al., 2015;Yu et al.,
2020). The suppression effect of $NO_3^-$ on the $N_2O_5$ heterogeneous uptake is mainly
caused by the competition of aerosol nitrate with chloride and $H_2O$ for the $H_2ONO_2^+$
intermediate (Bertram and Thornton, 2009). The positive correlation ($R^2=0.48$)
between $\gamma(N_2O_5)$ and molar ratio of $Cl^-/NO_3^-$ values was weaker than that of ALWC
(Figure 3d), which indicates that $Cl^-$ may promote the $N_2O_5$ uptake reaction instead of
playing a critical role during our observation. The particulate $Cl^-$ concentration also
contributes to weaker enhancement of $\gamma(N_2O_5)$ compared to ALWC in other field
observations (Wang et al., 2020b;Yu et al., 2020;McDuffie et al., 2018).



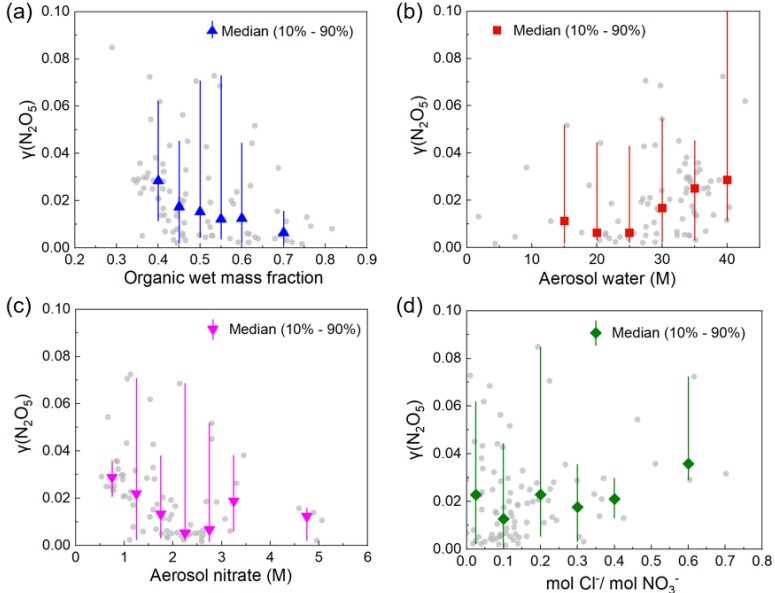

**Figure 3. The functional dependence of measured γ(N₂O₅) values on the influencing factors.** Variation of γ(N₂O₅) with organic wet mass fraction (a), the aerosol water content (b), the aerosol nitrate content (c), and molar ratio of chloride to nitrate (d). The points represent the median in each bin, and the color lines represent the data range from 10th to 90th percentile in each bin.

**3.3. Comparison of parameterized γ(N₂O₅) values**

The γ(N₂O₅) values were predicted using ten widely-used parameterizations and compared with the measured results. The details of the parameterizations were summarized in Table S3. Parameterizations were categorized into inorganic-only and inorganic kernel with organic coating or organic mass (inorganic+organic).

The γ(N₂O₅) predicted by inorganic-only parameterizations were generally larger than measurements. Among these inorganic-only parameterizations, RIE03, BT09 w/o Cl and Yu20 exhibited relatively low deviation in predicted median values from measurements (Figure 4a). However, the correlation of predictions and measurements were bad for these three parameterizations ($R^2$=0~0.09, Figure 4b). The empirical parameterization Yu20 derived from several field campaigns in China showed the best performance with a median difference of 4%, the lowest RMSE (0.0200) and the highest correlation coefficient ($R^2$=0.09) in Kunming, indicating the effectiveness of the improvement by the localized field results. The overestimation of the DAV08, BT09



and GRI09 were also reported by previous studies (Bertram et al., 2009b;Brown et al.,
2009;Chang et al., 2016;Griffiths et al., 2009;McDuffie et al., 2018). All
parameterizations had difficulty in predicting the low and high values of measured
$\gamma(N_2O_5)$. For the parameterizations with median deviation less than 10%, the
parameterized $\gamma(N_2O_5)$ values mainly fell in the range of 0.0036~0.035, while the
measured values varied from 0.0018 to 0.12, indicating that the relevant parameters in
the parameterizations was still inappropriate and cannot reproduce the range of the
measurements.

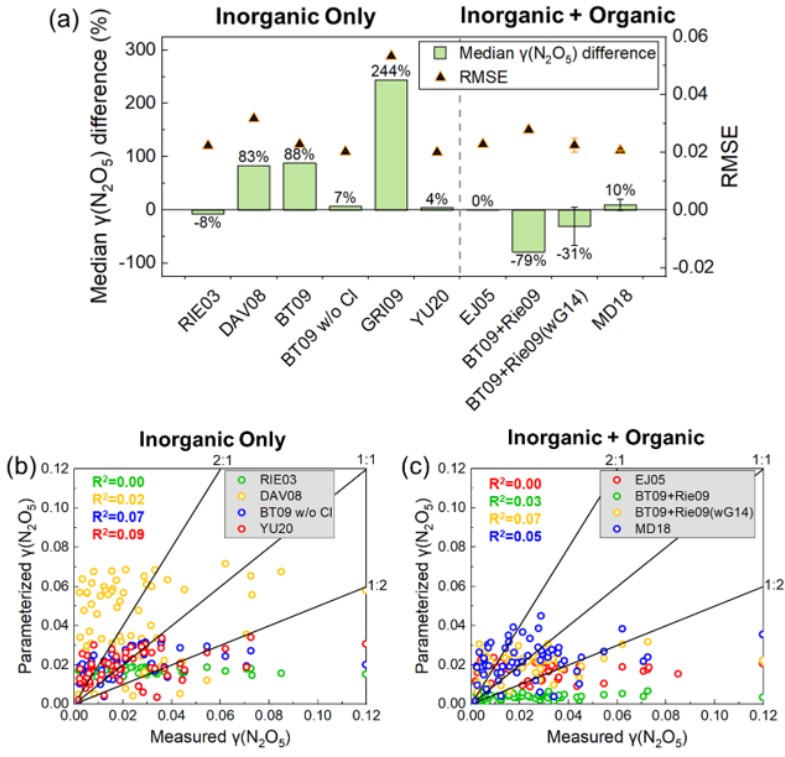


**Figure 4. Comparison of parameterized and measured $\gamma(N_2O_5)$.** (a) Comparison of
median difference and root-mean-square-error (RMSE) between measured $\gamma(N_2O_5)$ and
parameterized $\gamma(N_2O_5)$ values. The error bar of BT09+Rie09wG14 and MD18 showed
the range of the results of O/C setting between 0.5 and 0.8. The distribution of
parameterized $\gamma(N_2O_5)$ values, including inorganic-only parameterizations (RIE03,
DAV08, BT09 w/o Cl[-], YU20) (b), and inorganic+organic parameterizations (EJ05,
BT09+Rie09, BT09+Rie09wG14, MD18) (O/C=0.8) (c). The black solid line
represents the 1:1, 1:2 and 2:1 line, respectively. The $R^2$ displayed in different colors



correspond to the parameterization of the same color.

The inorganic+organic parameterizations tend to underestimate the measured

$\gamma(N_2O_5)$ due to the suppression effects of organics. Worse agreement and larger scatter
were found for the parameterized $\gamma(N_2O_5)$ ($R^2$=0~0.07, Figure 4c) when organics part
was added into inorganic. BT09+Rie09(wG14) showed the best correlation with $R^2$ of
0.07 but relatively large median deviation (-66~5%). EJ05 and MD18 showed the
lowest deviations among the four parameterizations, while EJ05 showed the worst
correlation ($R^2$=0.00). Among them, the empirical parameterization MD18, derived
from field observations, exhibited the best performance with a deviation of -1~20% and
the lowest RMSE (0.0207), which also indicates that parameterization can be improved
by fitting to field observations, similar to the results of inorganic-only
parameterizations.
**3.4. The impact of ALWC on parameterized $\gamma(N_2O_5)$**

Although some parameterizations performed relatively well in reproducing the

median values of $\gamma(N_2O_5)$, none of the ten parameterizations were able to reproduce the
range of measured $\gamma(N_2O_5)$ values (aka. low correlation and RMSE). This phenomenon
was possibly caused by several aspects, including the inaccurate estimation on response
coefficients of critical aerosol compositions and relative rates of competitive reactions,
especially when influenced by organics components. ALWC is one of the factors
controlling $N_2O_5$ uptake during our observation and the coefficients related to ALWC
should play a critical role in reproducing the varying range of $\gamma(N_2O_5)$. To investigate
the accuracy of the ALWC-related response coefficients in $\gamma(N_2O_5)$ parameterizations,
we compared the parameterized and measured $\gamma(N_2O_5)$ values at three ALWC levels:
low concentration (0~25 M), medium concentration (25~35 M), and high concentration
(35~45 M).

Six parameterizations were selected for the comparison at different ALWC levels

due to their low deviations (below 10% of median values) over the entire observation
(Figure 5). At low ALWC, all six parameterizations showed overestimation with the
maximum difference for EJ05 (189%) and the minimum for MD18 (34%). At median
ALWC, the deviation of parameterized $\gamma(N_2O_5)$ reduced to -8~4%. At high ALWC, the
parameterizations tend to underestimate the measured $\gamma(N_2O_5)$ with the difference
ranging from -37% to -1%. The treatment of ALWC-related effects on the $\gamma(N_2O_5)$
following BT09 and Rie09 parameterizations framework were generally better than
those following RIE03 and EJ05. The YU20 and MD18 showed the best performance



across all three ALWC levels among inorganic-only parameterizations and
inorganic+organic parameterizations, respectively. As a result, the overestimation at
low ALWC and underestimation at high ALWC indicated that treatments of ALWC-
related coefficients in most parameterizations can hardly reproduce the response of
$\gamma(N_2O_5)$ to ALWC.

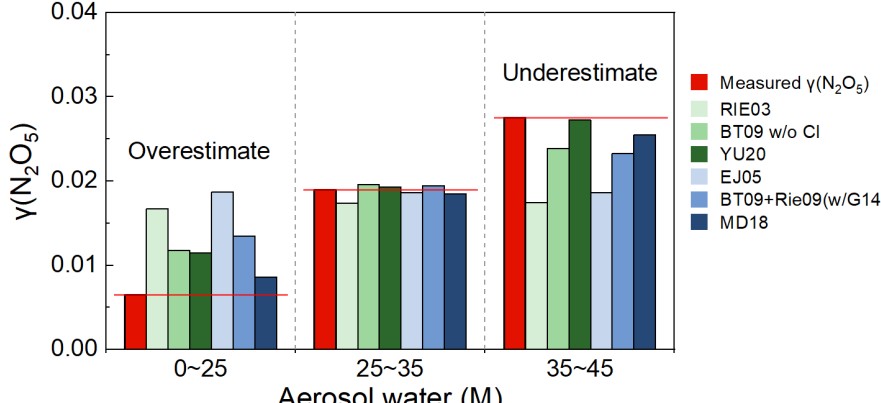


**Figure 5. Comparison of the median values of measured and parameterized**
**$\gamma(N_2O_5)$ at low, median and high ALWC levels.** The O/C settings of
BT09+Rie09wG14 and MD18 were 0.8 and 0.5, respectively.

The biased prediction of $\gamma(N_2O_5)$ at low and high ALWC levels might cause
considerable uncertainties on estimating impacts of $N_2O_5$ uptake when ALWC varies
largely in ambient conditions. We calculated the particulate nitrate production potential
contributed by $N_2O_5$ uptake based on measured $\gamma(N_2O_5)$ and six selected
parameterizations at low and high ALWC levels, respectively. The maximum deviations
of median nitrate production rates were 74% and -77% at low and high ALWC levels,
respectively (Figure 6). Our results indicate that current parameterizations may lead to
large deviations of nitrate production potential predictions. The contribution of the
$N_2O_5$ heterogeneous reaction to nitrate production is important in some regions (Wang
et al., 2021;Chen et al., 2020a;Fan et al., 2020;Wagner et al., 2013), and can be
comparable with that of OH+NO$_2$ pathway (Alexander et al., 2020;Fan et al., 2022;Zhai
et al., 2023). Therefore, we suggest more $\gamma(N_2O_5)$ measurements need to be conducted
under extreme ALWC conditions in future studies, which helps to improve the accuracy
of response coefficients.



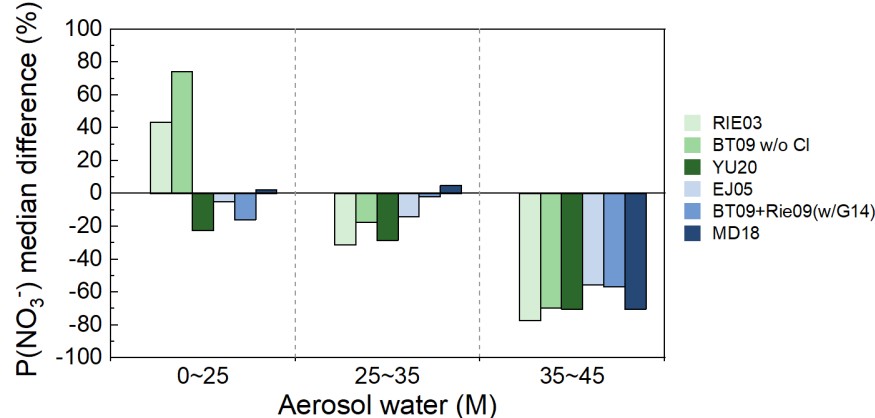


**Figure 6. The median difference of the nitrate production rates P(NO₃⁻) between measured and parameterized γ(N₂O₅) values during low, median and high ALWC conditions.** The O/C setting of BT09+Rie09wG14 was 0.8 and that of MD18 was 0.5.

## 4. Conclusions

The $\gamma(N_2O_5)$ on ambient aerosols were directly measured in Kunming by an aerosol flow tube system. The observed values showed good correlations with organics, ALWC and aerosol nitrate. The median of $\gamma(N_2O_5)$ predicted by inorganic-only and inorganic+organic parameterizations generally overestimate and underestimate the measurements, respectively. While some parameterizations agreed well with the measurements on median values, they failed to reproduce the varying range and showed low correlations. In particular, parameterizations overestimate the $\gamma(N_2O_5)$ by 34~189% at low ALWC and underestimate by -37%~-1% at high ALWC, respectively. Among the ten parameterizations, the empirical parameterizations YU20 and MD18 performed relatively well with lower deviations on median values and RMSE. Our result uncovers the feasibility of fitting with ambient measurements to improve laboratory-derived parameterizations. Therefore, we call for the need to conduct more field observations of $\gamma(N_2O_5)$ directly on ambient aerosols to improve the performance of parameterizations and better elucidate the environmental impacts of $N_2O_5$ uptake reaction. Meanwhile, further studies on mechanism of $N_2O_5$ uptake under extreme ALWC conditions would help to improve the accuracy of its response coefficients in parameterizations.

402



## Author information

**Corresponding Authors**

**Keding Lu** – State Key Joint Laboratory of Environmental Simulation and Pollution Control, College of Environmental Science and Engineering, Peking University, Beijing 100871, China; Email: k.lu@pku.edu.cn

**Xiaorui Chen** – School of Atmospheric Sciences, Sun Yat-sen University, and Southern Marine Science and Engineering Guangdong Laboratory (Zhuhai), Zhuhai, 519082, China; Email: chenxr95@mail.sysu.edu.cn

**Authors**

**Jiayin Li**[⊥] – State Key Joint Laboratory of Environmental Simulation and Pollution Control, College of Environmental Science and Engineering, Peking University, Beijing 100871, China

**Tianyu Zhai**[⊥] – State Key Joint Laboratory of Environmental Simulation and Pollution Control, College of Environmental Science and Engineering, Peking University, Beijing 100871, China; State Environmental Protection Key Laboratory of Vehicle Emission Control and Simulation, Chinese Research Academy of Environmental Sciences, Beijing, 100012, China

**Haichao Wang** – School of Atmospheric Sciences, Sun Yat-sen University, and Southern Marine Science and Engineering Guangdong Laboratory (Zhuhai), Zhuhai, 519082, China

**Shuyang Xie** – State Key Joint Laboratory of Environmental Simulation and Pollution Control, College of Environmental Science and Engineering, Peking University, Beijing 100871, China

**Shiyi Chen**– State Key Joint Laboratory of Environmental Simulation and Pollution Control, College of Environmental Science and Engineering, Peking University, Beijing 100871, China

**Chunmeng Li**– State Key Joint Laboratory of Environmental Simulation and Pollution Control, College of Environmental Science and Engineering, Peking



University, Beijing 100871, China; The National Institute of Metrology, Center for
Environmental Metrology, Beijing 100029, China
**Huabin Dong**– State Key Joint Laboratory of Environmental Simulation and
Pollution Control, College of Environmental Science and Engineering, Peking
University, Beijing 100871, China

## Author Contributions

X.R.C. and K.D.L. designed the study. K.D.L organized the field campaign with the
help from Y.J.G. T.Y.Z and X.R.C measured the $\gamma(N_2O_5)$ data. C.M.L, S.Y.X, H.B.D
and S.Y.C provide the field data of normal gases, particulate components and other
supporting parameters. J.Y.L, T.Y.Z, X.R.C and H.C.W analyze the data. J.Y.L, T.Y.Z
and X.R.C wrote the paper with the input from K.D.L.

## Competing Interests

The authors declare no competing financial interest.

## Acknowledgments

This work was supported by the National Natural Science Foundation of China
(Grants No. 22221004 and No. 22406204).

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
