# Peer review of "Direct measurement of N2O5 heterogeneous uptake"

_EGUsphere, 2024_

## Referee Comment (RC1)

Overall comments:

This work performed direct measurements of $N_2O_5$ heterogeneous uptake coefficients on atmospheric aerosols in southwestern China and further compared the measured results with those predicted based on different parameterization scenarios. Considering that most of previous studies focusing on gas uptake kinetics are conducted under laboratory conditions, the obtained uptake coefficients may be deviate from the real case in the ambient air. The authors in this work made a step forward and quantified the uptake coefficients of $N_2O_5$ on real ambient aerosols. The results will help to constrain the uptake coefficients of $N_2O_5$ to a more realistic basis and further improve our understanding of the heterogeneous reaction kinetics under ambient conditions and their potential impacts on aerosol formation at least locally. Generally, this work could be a meaningful addition into the literature. However, there are several major issues (as pointed out below) need to be addressed at the present stage.

Major issues:

Regarding the methods, the related descriptions on the experiment using the aerosol flow tube system are missing key messages. Please give a more detailed description about the air sampling system. Did ambient air directly enter into the flow tube? The measurement system can generate $N_2O_5$ by itself, but how did you deal with the $N_2O_5$ in the ambient air? This would also influence the obtained uptake coefficient. For uptake coefficient calculation, the authors took the flow tube wall effect into account, how did the authors consider the effect caused by aerosol wall losses? Did the wall loss also apply for the gas and particles? What uncertainties would the authors expect for the obtained uptake coefficients?

Even though for some cases the parameterized gamma agreed well with the measured gamma (median values), the correlations between them were very bad for all the parameterization scenarios. It would be more helpful to have additional discussions regarding this. Is this more likely caused by parameterization methods or the measurement method? More suggestions on how to choose the different parameterization scenarios under various conditions would be more meaningful from the modelers' point of view.

The presentation quality of manuscript seems to be poor. The use of the language appears to be a big problem. Some mistakes should have been avoided if the authors carefully inspect the text before the submission. As shown below, additional edits need to be done regarding the "low level" grammar mistakes. Please note that these grammar issues are not limited to the following list. The authors should therefore check through the whole manuscript very carefully for the revised version.

Minor suggestions/edits:

Line 41-42: What does "response coefficients" mean?

Line 91-95: "However, the comparisons of …." Please rephrase this sentence, as it reads unclearly and awkwardly.

Line 96: "can leads to" should be "can lead to".

Line 105: "European" should be "Europe".

Line 113-114: "We further notice …" the whole sentence reads awkwardly. Please rephrase it.

Line 126: What does CNST mean?

Line 129: "Google Maps images" should it be "Google Map images" ?

Line 138: "and" should be "which".

Line 147: "A total of 117 kinds of VOCs …" could be like "A total of 117 VOCs species …".

Line 155: "included" should be "including".

Line 158: For the header of Table 1, "Detection of limit" should be "Detection limits", "Method" should be "Methods", "Accuracy" should be "Accuracies".

Line 160: "The" is not needed in front of "measurement", for this case. The same applies to the title of the other sections.

Line 165: "(excess)" should be "(in excess)".

Line 166: "consist of" should be "consists of".

Line 170: "concentration" should be "concentrations".

Line 169-176: These three sentences read awkwardly and apparently have several grammatical mistakes. Please rewrite them.

Line 192: "the reaction rate of constant of…" should be "the reaction rate constant of …".

Line 209: "The mean diurnal of measured…" should be "The mean diurnal variation of the measured …". The authors seem to have a big problem on how to correctly use "the".

Line 216: "(PNO3)" should be "P(NO3)".

Line 218: "observation campaign" is not the common way. Either "observation period" or just "campaign".

Line 252: "Functional dependence" reads very awkwardly. Please change it.

Line 254: What is "organic wet mass faction"? How did you measure it?

Line 260-262: As the authors stated, "the negative correlation of particulate organic and $\gamma(N2O5)$ was usually weak derived from field measurements". In the present work, however, the authors found a significant negative correlation between the organic wet mass fraction and the gamma. Could the authors discuss why this study shows such a difference?

Line 275: "showed" should be "shows".

Line 290: In the caption of Fig. 3, the authors state that "The points represent the median in each bin,…". How do the authors select the different bins? Should be the symbols or the points represent the median values?

Line 338: "(aka. low …)", for me it is the first time to see the use of "aka.". Probably it is not the official way to use it in a scientific research paper?

Line 339-340: The authors use "response coefficients" several times throughout the manuscript. At least it is not a scientifically meaningful definition, as far as I know. What is the exact meaning of this? Any references?

Line 358-361: Please rephrase the whole sentence to make it more scientifically readable.

Line 377-379: Same as above.

---

## Author Comment (AC1)

**Reply on comments**

We appreciate the reviewer for the careful reading and their constructive comments on our manuscript. **As detailed below, the reviewer's comments are** in normal font, **and our responses to the comments are shown in** *italicized font*. New or modified text is in blue.

All the line numbers refer to the original version of Manuscript ID: **egusphere-2024-3804**

The authors directly measured the uptake coefficient of $N_2O_5$ using an aerosol flow tube reactor in Kumin, a highland environment, in China. The possible influence factors on the $\gamma(N_2O_5)$ were discussed by correlation analysis in this work. ALWC was the most important one affecting the measured $\gamma(N_2O_5)$. The comparison between the measured $\gamma(N_2O_5)$ and parameterized values. Overall, the results are well presented and discussed. It is publishable after the following questions have been well addressed.

1. Considering many studies on the $\gamma(N_2O_5)$ by measurement, modeling, and parameterization methods, I am wondering what scientific assumption is focused on in this work. The uptake kinetics in different environments? Crucial factors determining the $\gamma(N_2O_5)$? Or the uncertainties of the parameterization method for $\gamma(N_2O_5)$? The authors should clarify the highlights and the novelty of this study.

*Thanks for the suggestions. We mainly focused on the main factors determining the $\gamma(N_2O_5)$ in a southwestern area of China with high sea level and low influence from industrial activities. No $\gamma(N_2O_5)$ observation has been reported in this region, which therefore provide us an opportunity to evaluate the general performance of current $\gamma(N_2O_5)$ parameterizations under this typical environment. We found that median $\gamma(N_2O_5)$ predicted by some parameterizations agreed well with observations, whereas the parameterizations failed to reproduce the range of observed values and showed poor correlations. We found significant discrepancies between observed and*

*parameterized values at extreme levels of aerosol water content, indicating the need for the improvement of the response coefficients of aerosol water content. In addition, the relationship between the measured γ(N₂O₅) values and impacting factors was consistent with previous laboratory results, except for aerosol chloride.*

*This study introduces several novel contributions: First, the γ(N₂O₅) values were quantified through direct measurements with high accuracy and reliability. Second, a significant relationship between key influencing factors and the measured values was identified, providing deeper insights into the underlying mechanisms. Finally, this study systematically analyzed the possible reasons of discrepancies between parameterized and measured values, offering an evaluation for existing models. These advancements enhance our understanding of N₂O₅ uptake dynamics and provide a new insight for improving parameterization performances.*

According to the above discussion, we revised the abstract as follows.

"The heterogeneous hydrolysis of dinitrogen pentoxide ($N_2O_5$) is a critical process in assessing $NO_x$ fate and secondary pollutants (e.g. particulate nitrate) formation. However, accurate quantification of the $N_2O_5$ uptake coefficient ($\gamma(N_2O_5)$) in ambient conditions is a challenging problem that can cause unpredictable uncertainties in the predictions of the air quality models. Here, the $\gamma(N_2O_5)$ values were directly measured using an improved in situ aerosol flow tube system at a site located on a highland region in southwestern China to investigate its influencing factors and performances of current $\gamma(N_2O_5)$ parameterizations under this typical environmental condition. The nocturnal mean $\gamma(N_2O_5)$ value ranged from 0.0018 to 0.12 with an average of 0.023±0.021. The relationship between the measured $\gamma(N_2O_5)$ values and impacting factors was consistent with previous laboratory results, except for aerosol chloride. The aerosol water significantly promoted $N_2O_5$ uptake, while particulate organic and nitrate showed suppression effects. We found that several parameterizations can capture the median value of measured values, whereas none of the ten parameterizations were able to reproduce the variabilities and showed poor correlations ($R^2$=0.00~0.09). Elevated biases of predictions specifically occurred at high aerosol liquid water content (ALWC) (>35M) and low ALWC (<25M) with an underestimation of -37%~-1% and an

overestimation of 34~189%, respectively. Such differences between the measured and parameterized $\gamma(N_2O_5)$ would lead to biased estimation (-77%~74%) on particulate nitrate production potential. Our findings suggest the need for more direct field quantifications of $\gamma(N_2O_5)$ and the laboratory measurements under extreme ALWC conditions to re-evaluate the response coefficients between $\gamma(N_2O_5)$ and aerosol chemical compositions in parameterizations. "

And we revised the conclusion as follows.

"The $\gamma(N_2O_5)$ values on ambient aerosols were directly measured by an improved in situ aerosol flow tube system in Kunming, which represents a typical highland environment. The relationship between the measured $\gamma(N_2O_5)$ and impacting factors was consistent with previous laboratory results, except for aerosol chloride. The median of $\gamma(N_2O_5)$ predicted by inorganic-only and inorganic+organic parameterizations generally overestimate and underestimate the measurements, respectively. While some parameterizations agreed well with the measurements on median values, they failed to reproduce the variabilities and showed low correlations. In particular, parameterizations overestimate the $\gamma(N_2O_5)$ by 34~189% at low ALWC and underestimate by -37%~-1% at high ALWC, respectively. Among the ten parameterizations, the empirical parameterizations YU20 and MD18 performed relatively well with lower deviations in median values and RMSE. The suggestions on how to choose the different parameterization scenarios under various conditions were given. Our result reveals that using ambient measurements can effectively improve parameterizations derived from laboratory experiments. Therefore, we call for the need to conduct more field observations of $\gamma(N_2O_5)$ directly on ambient aerosols to improve the performance of parameterizations and better elucidate the environmental impacts of $N_2O_5$ uptake reaction. Meanwhile, further studies on the mechanism of $N_2O_5$ uptake under extreme ALWC conditions would help to improve the accuracy of its response coefficients in parameterizations."

2. It is somewhat too weak about the correlation analysis between the measured $\gamma(N_2O_5)$ and possible factors such as chemical composition and ALWC. The dataset is too small

and the data points are too scattering. So, it is hard to observe the significant correlation.

*Thanks for the valuable comments and we understand the reviewer's concern. To ensure accurate measurements, a rigorous data screening process was implemented. A 10% cutoff for $N_2O_5$ variation was applied to excluding air masses that were too unstable for valid analysis according to our data screening criteria. Cases showing more than a 2% variation in relative humidity (RH) between HEPA inline and bypass modes were excluded due to significant influence of RH on $k_{wall}$ of $N_2O_5$ wihthin the aerosol flow tube. To ensure significant $N_2O_5$ concentration differences resulted from heterogeneous uptake reactions between the top and bottom of the flow tube, periods with low Sa conditions ($<100 \ \mu m^2 \ cm^{-3}$) were filtered out. Additionally, cases where NO concentration exceeded 7 ppbv were excluded to avoid significant changes in $NO_3$-$N_2O_5$ concentration due to NO titration in the flow tube. Therefore, we get a relatively small dataset during this 35-day observation.*

*The correlations between the measured values and impacting factors seem not so significant within this ambient dataset. Actually, $\gamma(N_2O_5)$ in a real atmosphere can be affected by a combination of factors, which results in lower correlations of them than laboratory studies. Even so, the correlation ($R^2$) between aerosol liquid water and $\gamma(N_2O_5)$ in this study was 0.74, which is still higher than previous studies in China (0.65) and USA (0.15)(Yu et al., 2020;McDuffie et al., 2018;Wang et al., 2020) (Figure R1). To further confirm the relationships, we divide each impacting factor into three groups from smallest to largest. The trends of $\gamma(N_2O_5)$ varying with factors are shown in Figure R2. We find that $\gamma(N_2O_5)$ decreases significantly with organic wet mass fraction and aerosol nitrate, and increases with aerosol liquid water as well as $Cl^-/NO_3^-$, which are consistent with previous laboratory studies.*

[Figure]

***Figure R1. The correlations between γ(N₂O₅) and impacting factors in different field studies.*** *a:five field campaigns in China(Yu et al., 2020), b: field campaign in Beijing, China(Wang et al., 2020), c: field campaign in USA(McDuffie et al., 2018).*

[Figure]

***Figure R2. The correlations between measured γ(N₂O₅) values and impacting factors.*** *Variation of γ(N₂O₅) with organic wet mass fraction (a), the aerosol water*

*content (b), the aerosol nitrate content (c), and molar ratio of chloride to nitrate (d).*
*Each impacting factor was divided into three groups, the error bars represent the data*
*range from 10th to 90th percentile in each bin.*

3. About the negative correlation between organic aerosol and $\gamma(N_2O_5)$, is the $\gamma(N_2O_5)$ possibly affected by other factors when the concentration of organic is high along with a low $\gamma(N_2O_5)$? Unless organic aerosol is well mixed with other components, i.e., in an ideal inner mixed state.

*Thanks for your comments. The negative correlation between organic wet mass fraction and $\gamma(N_2O_5)$ actually may not be influenced by other factors due to the possible formation of organic coating on aerosols when the concentration of organic is high in this study. Previous laboratory studies have proved that organics can suppress the uptake reaction by forming a coating on the surface of the particles and regulating the aerosol liquid water content and the passage rate of $N_2O_5$ molecules into particles (Folkers et al., 2003;Gaston et al., 2014;Anttila et al., 2006)). To investigate this problem, a table of Pearson Correlation Coefficient (r) among factors affecting $\gamma(N_2O_5)$ was calculated, as shown in Table S3, to find the covariations of organics and other aerosol components or impact factors. The organic wet mass fraction showed a negative correlation with aerosol water (r=-0.79), and the correlations with other factors were insignificant. It indicates that the organic coating may occur on most aerosols and limit the penetration of the liquid water into the bulk phase. Therefore, organics might affect $\gamma(N_2O_5)$ mainly by regulating aerosol liquid water content and thus limit the effects of other factors in this study.*
*We added the discussion in section 3.2 in the main text as follows.*

"The organic wet mass fraction in this study varies between 0.3 and 0.8, while other previous studies have reported a variation range of 0.1 to 0.5(McDuffie et al., 2018;Wang et al., 2020;Brown et al., 2009). The large proportion and variation range of organics in the aerosols may lead to a more significant inhibition effect on $\gamma(N_2O_5)$. Additionally, we found that both the dry and wet mass fractions of organics in this study showed significant negative correlations with ALWC, with Pearson coefficients of -

0.66 and -0.79 (Table S3), respectively. Therefore, organics might decrease $\gamma(N_2O_5)$ by forming an organic coating to limit the penetration of liquid water into the particle phase and hinder the reaction between $N_2O_5$ with liquid phase."

*The table added in supporting information is as follows.*

**Table S3. Pearson Correlation Coefficient (r) among factors affecting $\gamma(N_2O_5)$, the statistical data are limited to the period in which measured $\gamma(N_2O_5)$ is available.**

| Factors | Temp | RH | Aerosol Water | Aerosol Nitrate | $H_2O/NO_3^-$ Molar ratio | Aerosol Chloride | $Cl^-/NO_3^-$ Molar ratio | Org dry mass fraction | Org wet mass fraction | $Org/SO_4^{2-}$ |
|---|---|---|---|---|---|---|---|---|---|---|
| Temp | 1.00 | -0.85 | -0.45 | -0.27 | 0.09 | -0.30 | -0.12 | 0.15 | 0.24 | 0.09 |
| RH | - | 1.00 | 0.49 | 0.45 | -0.27 | 0.24 | -0.09 | -0.12 | -0.23 | -0.08 |
| Aerosol Water | - | - | 1.00 | -0.40 | 0.62 | -0.10 | 0.19 | -0.66 | -0.79 | -0.68 |
| Aerosol Nitrate | - | - | - | 1.00 | -0.80 | 0.24 | -0.39 | 0.34 | 0.38 | 0.43 |
| $H_2O/NO_3^-$ Molar ratio | - | - | - | - | 1.00 | -0.21 | 0.45 | -0.64 | -0.67 | -0.67 |
| Aerosol Chloride | - | - | - | - | - | 1.00 | 0.69 | -0.02 | 0.01 | 0.06 |
| $Cl^-/NO_3^-$ Molar ratio | - | - | - | - | - | - | 1.00 | -0.37 | -0.35 | -0.33 |
| Org dry mass fraction | - | - | - | - | - | - | - | 1.00 | 0.98 | 0.97 |
| Org wet mass fraction | - | - | - | - | - | - | - | - | 1.00 | 0.96 |
| $Org/SO_4^{2-}$ | - | - | - | - | - | - | - | - | - | 1.00 |

4. ALWC is the most important factor affecting $\gamma(N_2O_5)$. However, ALWC is greatly determined by nitrate concentration or fraction in PM2.5. Why does $\gamma(N_2O_5)$ show a different dependence on nitrate like ALWC?

*ALWC can be affected by many factors, such as RH, particle sulfate, particle nitrate, and particle organics. In this study, as shown in Table S3, ALWC mainly correlates with organic dry mass fraction (r=-0.66) and RH (r=0.49). The relatively low nitrate mass fraction (5%) observed in Kunming suggests its potential contribution*

*to ALWC may be limited in this region.*

*For the reason of different dependence, particulate nitrate acts as a competing role to $H_2O$ when participates in $N_2O_5$ uptake reaction in the bulk phase. The currently proposed mechanism for the $N_2O_5$ heterogeneous uptake (R1-R4) posits that upon entering the liquid phase, $N_2O_5$ initially decomposes into nitrate ($NO_3^-$) and the intermediate ion $H_2ONO_2^+$. Reactions R1 and R2 are reversible, and an increase in $NO_3^-$ concentration inhibits the decomposition of $N_2O_5$ in the liquid phase, promoting its desorption back into the gas phase. Hence, particulate nitrate and $\gamma(N_2O_5)$ showed a negative correlation which is different from ALWC.*

$N_2O_5 (g) \leftrightarrow N_2O_5 (aq)$                                                 *(R1)*

$N_2O_5(aq) + H_2O(l) \leftrightarrow H_2ONO_2^+(aq) + NO_3^- (aq)$            *(R2)*

$H_2ONO_2^+ (aq) + H_2O(l) \rightarrow HNO_3(aq) + H_3O^+ (aq)$         *(R3)*

$H_2ONO_2^+ (aq) + HX(aq) \rightarrow XNO_2(aq) + H_3O^+ (aq)$           *(R4)*

*We revised the discussion in section 3.2 to further clarify this point in the main text as follows.*

"The suppression effect of $NO_3^-$ on the $N_2O_5$ heterogeneous uptake is mainly caused by the competition of aerosol nitrate with chloride and $H_2O$ for the $H_2ONO_2^+$ intermediate (R5~R8) (Bertram and Thornton, 2009)."

5. In Figure 2b, it is better to compare the $\gamma(N_2O_5)$ distribution among different methods.

*According to the reviewer's suggestion, we attempted to calculate the uptake coefficient using indirect quantitative methods. However, previous studies have demonstrated that indirect quantitative methods have strict applicable scenarios. The steady-state method is more suitable for air masses with high aerosol concentrations, high temperatures, and moderate levels of $kNO_3$, making it more applicable in polluted regions with high aerosol loading during summertime(Chen et al., 2022). The box model method is significantly affected by uncertainties of $N_2O_5$ concentration and may cause order-of-magnitude overestimations under conditions of low air mass age, low $O_3$ concentration or high NO concentration, and high $N_2O_5$ uptake(Wagner et al.,*

*2013;Chen et al., 2022). During our observation, the measurements were influenced by NO concentration from traffic emissions. The air mass was relatively clean with low $N_2O_5$ concentration, which did not satisfy the necessary conditions for applying the iterative box model and steady-state method to calculate $\gamma(N_2O_5)$. Therefore, it is not feasible to obtain accurate $\gamma(N_2O_5)$ values using indirect quantitative methods in this study.*

6. In Figure 4, the performances of these parameterization methods look too bad. I am wondering how about the performances of models widely used in literature.

*The widely used parameterizations in models mainly include BT09, BT09w/oCl and MD18. The BT09 always overestimates the median $\gamma(N_2O_5)$ values, primarily attributable to its overestimated enhancement effect of chloride (Cl-)(Chang et al., 2016;Yu et al., 2020;Morgan et al., 2015). Similar results have also been found in this study with an 88% overestimation. In this study, BT09w/oCl and MD18 capture the median well, but fail to reproduce the range and variability of the measured values. Although these two parameterizations show less than 10% deviation compared to measured values, they exhibit notably low correlation coefficients ($r^2 = 0.07$ for BT09w/oCl and 0.05 for MD18). The low correlation has also been reported in the campaign in the United States ($R^2 = 0.18$ for BT09w/oCl and 0.17 for MD18)(McDuffie et al., 2018). In other regions, BT09w/oCl even showed different degrees of overestimations due to the absence of organic suppression or other influencing factors (Morgan et al., 2015;McDuffie et al., 2018). For instance, this parameterization overpredicts the median by 81% in Beijing, China (Wang et al., 2020).*

**References**

[revised manuscript text omitted]

---

## Author Comment (AC2)

**Reply on comments**

We appreciate the reviewer for the careful reading and their constructive comments on our manuscript. **As detailed below, the reviewer's comments are** in normal font, **and our responses to the comments are shown in** *italicized font*. New or modified text is in blue.

All the line numbers refer to the original version of Manuscript ID: **egusphere-2024-3804**

Overall comments:

This work performed direct measurements of $N_2O_5$ heterogeneous uptake coefficients on atmospheric aerosols in southwestern China and further compared the measured results with those predicted based on different parameterization scenarios. Considering that most of previous studies focusing on gas uptake kinetics are conducted under laboratory conditions, the obtained uptake coefficients may be deviate from the real case in the ambient air. The authors in this work made a step forward and quantified the uptake coefficients of $N_2O_5$ on real ambient aerosols. The results will help to constrain the uptake coefficients of $N_2O_5$ to a more realistic basis and further improve our understanding of the heterogeneous reaction kinetics under ambient conditions and their potential impacts on aerosol formation at least locally. Generally, this work could be a meaningful addition into the literature. However, there are several major issues (as pointed out below) need to be addressed at the present stage.

Major issues:

1. Regarding the methods, the related descriptions on the experiment using the aerosol flow tube system are missing key messages. Please give a more detailed description about the air sampling system. Did ambient air directly enter into the flow tube? The measurement system can generate $N_2O_5$ by itself, but how did you deal with the $N_2O_5$ in the ambient air? This would also influence the obtained uptake coefficient. For

uptake coefficient calculation, the authors took the flow tube wall effect into account, how did the authors consider the effect caused by aerosol wall losses? Did the wall loss also apply for the gas and particles? What uncertainties would the authors expect for the obtained uptake coefficients?

*Thanks for your valuable suggestions. In this aerosol flow tube system, ambient air is directly introduced into the system through a stainless steel sampling tube which removes the ambient $N_2O_5$ efficiently. The sampling air then mixes with $N_2O_5$ generated from an $N_2O_5$ source before entering the aerosol flow tube. After the mixing of ambient air and $N_2O_5$ source, the total concentration of $N_2O_5$ at the top of the flow tube is quantified by a cavity-enhanced absorption spectrometer (CEAS-PKU) before the gas enters the flow tube. This, combined with the measured $N_2O_5$ concentration at the bottom of the flow tube and other parameters, is used to calculate $\gamma(N_2O_5)$. The wall of the flow tube indeed causes the wall loss of aerosols. In previous experiments, we measured the loss of size-resolved Sa at both the top and bottom of the flow tube and determined the loss coefficient to correct for this loss. The total Sa loss caused by the flow tube is approximately 5%. The effect due to wall loss of $N_2O_5$ gas in the flow tube is mitigated by calibrating $k_{wall}$ every 20 min. We added detailed descriptions and uncertainties of measured gamma via the aerosol flow tube system in Supplementary Information as follows.*

**"S1. Detailed description of the measurement and calculation of $\gamma(N_2O_5)$**

The Aerosol Flow Tube System (AFTS) can be divided into three main modules: the sampling control module, the reaction module, and the detection module. The sampling of the flow tube is facilitated by a vacuum pump located at the end of the system. In the sampling control module, ambient air is directly introduced into the reaction pathway. The sampling gas passes through a 1-in/2-out solenoid valve that directs the sample either through a HEPA filter to remove aerosols or bypasses it, thereby controlling the presence of aerosols in the reaction module. The sampling gas is then mixed with a high concentration of $N_2O_5$ generated from a $N_2O_5$ source before entering the reaction module. At the top of the reaction module, two stainless steel static

mixers are installed to ensure that the gas is thoroughly mixed. The aerosol flow tube is the primary site for $N_2O_5$ uptake reactions.

During detection, concentrations of $NO_x$ and $O_3$ are continuously measured at the top of the flow tube to facilitate subsequent simulation of gas-phase reactions within the flow tube using a box model. The measurement of $N_2O_5$ concentration is conducted through two separate 20-minute processes: one to determine the $N_2O_5$ loss rate in the absence of aerosols ($k_{wall}$) and another in the presence of aerosols ($k_{wall}+k_{aerosol}$). The only difference between the two processes is the presence or absence of aerosols. Each process includes two steps: measuring $N_2O_5$ concentrations at both the top and bottom of the flow tube, each step maintains 10 min. Throughout the measurement process, the aerosol surface area ($Sa$) is continuously measured at the bottom of the flow tube, followed by size-resolved $Sa$ correction based on previously determined particle loss coefficients (Chen et al., 2022).

By inputting the measured concentrations of $NO_x$, $O_3$, and $N_2O_5$ at the top of the flow tube under both aerosol-free and aerosol-present conditions into the box model, the $NO_3$-$N_2O_5$ chemical reactions and related gas-phase reactions in the flow tube are simulated until the model's output $N_2O_5$ concentration matches the measured value at the tube's bottom. This process yields $k_{wall}$ and $k_{wall}+k_{aerosol}$, from which the $N_2O_5$ loss rate on aerosols ($k_{aerosol}$) is derived by subtraction. The $\gamma(N_2O_5)$ can then be calculated using established formulas (EqS1).

$$k_{N2O5} = 0.25 \times Sa \times \gamma \times C \qquad\qquad\qquad (EqS1)$$

The uncertainty in $\gamma(N_2O_5)$ is relevant to the measurement uncertainties of each instrument and the rapid fluctuations of various parameters. To ensure accurate measurements, a rigorous data screening process was implemented. A 10% cutoff for $N_2O_5$ variation was applied to exclude air masses that were too unstable for valid analysis according to our data screening criteria. Cases showing more than a 2% variation in relative humidity (RH) between HEPA inline and bypass modes were excluded due to RH's significant influence on $k_{wall}$ of $N_2O_5$ in the aerosol flow tube. To ensure significant $N_2O_5$ concentration differences due to heterogeneous uptake reactions between the top and bottom of the flow tube, periods with low $Sa$ conditions

($<100$ μm² cm⁻³) were filtered out. Additionally, cases where NO concentration exceeded 7 ppbv were excluded to avoid significant changes in $NO_3$-$N_2O_5$ concentration due to NO titration in the flow tube.

Therefore, the system may introduce a 2% measurement bias in $\gamma(N_2O_5)$ due to $N_2O_5$ concentration fluctuations, a bias of $\pm8\times10^{-4}$ to $\pm2\times10^{-3}$ due to RH fluctuations, a 16% uncertainty from Sa measurement and particle loss in the flow tube, a 4% measurement fluctuation from Monte Carlo simulations, up to a 9% uncertainty from ambient temperature variations, and a 5% uncertainty from $NO_x$ and $O_3$ concentration fluctuations. In summary, considering all the factors and their corresponding varying ranges discussed above, the overall uncertainty of $\gamma(N_2O_5)$ determined from Monte Carlo simulations ranges from 16% to 43%.

[Figure]

**Figure S1.** Overall schematic of aerosol flow tube system. Bold arrows indicate the main lines of the sampling gas. ″

2. Even though for some cases the parameterized gamma agreed well with the measured gamma (median values), the correlations between them were very bad for all the parameterization scenarios. It would be more helpful to have additional discussions regarding this. Is this more likely caused by parameterization methods or the measurement method? More suggestions on how to choose the different parameterization scenarios under various conditions would be more meaningful from the modelers' point of view.

*Thanks for the suggestion. The poor correlation is mainly attributed to the response coefficients of impacting factors in current parameterizations, which failed to reproduce the observations at very high or low levels of these factors. Our analysis identified that inappropriate ALWC response coefficients in current parameterizations contributes to the bias of parameterizations. As shown in Figure 5 of the main text, at low ALWC, all six parameterizations showed overestimation with the maximum difference for EJ05 (189%) and the minimum for MD18 (34%). At median ALWC, the deviation of parameterized $\gamma(N_2O_5)$ reduced to -8~4%. At high ALWC, the parameterizations tend to underestimate the measured $\gamma(N_2O_5)$ with the difference ranging from -37% to -1%. Therefore, we recommend conducting kinetic experiments under extreme ALWC conditions to enhance the fitting efficacy of the parameterizations. A better performance of current parameterizations may also be realized by including parameters, such as particle morphology, phase state, and mixing state (You et al., 2014;Shiraiwa et al., 2017;Ng et al., 2010). These parameters, which are difficult to measure in field studies, have been demonstrated to affect $\gamma(N_2O_5)$. In previous research, McDuffie quantified $\gamma(N_2O_5)$ using a box model and also found poor agreements between the 14 parameterized and $\gamma(N_2O_5)$ values(McDuffie et al., 2018).*

*We added and revised the discussion in section 3.4 in the main text as follows:*

*"This phenomenon was possibly caused by several aspects, including the inaccurate estimation of response coefficients of aerosol compositions, relative rates constants of competitive reactions, and the missing parameters. The missing influencing factors in current parameterizations include parameters such as particle morphology, phase state, and mixing state (You et al., 2014;Shiraiwa et al., 2017;Ng et al., 2010). These parameters, which current methodologies are difficult to measure in field conditions, have been proven to affect $\gamma(N_2O_5)$, and can contribute to the discrepancy between parameterized and measured values."*

*We added some suggestions on how to choose the different parameterization scenarios under various conditions in section 3.3 in the main text as follows.*

*"The commonly used parameterizations mainly consist of inorganic and inorganic+organic framework, such as BT09w/oCl, YU20, and MD18. In this study,*

among all parameterizations, YU20 demonstrated the best performance, most likely because YU20 was optimized based on datasets observed in four rural regions in China. BT09w/oCl also performed well in this study, overestimating the median by only 7%. However, poor performances of BT09w/oCl were still reported in Pearl River Delta and North China Plain (Wang et al., 2022;Wang et al., 2020). Conversely, the BT09w/oCl performed well in Northwestern Europe, mainly because $\gamma(N_2O_5)$ in Europe is predominantly controlled by the ions in bulk phase (Morgan et al., 2015;Chen et al., 2018;Phillips et al., 2016). In North America, $\gamma(N_2O_5)$ is significantly inhibited by organic effects (Chang et al., 2016). The parameterizations considering organic effects, like MD18, might be more suitable for the conditions in North America. However, in this study, MD18 showed an overestimation of up to 20%, suggesting that this parameterization is not suitable for China, but more applicable to North American regions.

Hence, most regions in China, where $\gamma(N_2O_5)$ is controlled by aerosol liquid water content, are more suited to the YU20. European regions, where gamma is controlled by $H_2O/NO_3^-$ and less influenced by organics, are better served by the BT09w/oCl. Meanwhile, MD18 is more appropriate for North American regions. Localized parameterizations established on the basis of local measurements can exhibit superior performance within the respective regions. Parameterizations incorporating organic effects generally exhibit larger errors than others, underscoring the importance of further improving the consideration of organic effects in parameterizations."

3. The presentation quality of manuscript seems to be poor. The use of the language appears to be a big problem. Some mistakes should have been avoided if the authors carefully inspect the text before the submission. As shown below, additional edits need to be done regarding the "low level" grammar mistakes. Please note that these grammar issues are not limited to the following list. The authors should therefore check through the whole manuscript very carefully for the revised version.

Minor suggestions/edits:

Line 41-42: What does "response coefficients" mean?

*The response coefficients represent the quantitative relationship between $\gamma(N_2O_5)$ and aerosol chemical compositions in parameterizations. Such as $k_3$ and $k_{2b}$ in BT09 (Eq R1). It corresponds to the fitted relative rates of competing reactions(Yu et al., 2020), or represents functional relationships between aerosol chemical components and $\gamma(N_2O_5)$(McDuffie et al., 2018).*

$$\gamma = \frac{4}{c} \frac{V_a}{S_a} K_H k'_{2f} \left( 1 - \frac{1}{\left(\frac{k_3[H_2O]}{k_{2b}[NO_3^-]}\right) + 1 + \left(\frac{k_4[Cl^-]}{k_{2b}[NO_3^-]}\right)} \right) \qquad \text{Eq R1}$$

where,

$K_H$=51, Henry's Law Coefficient (Fried et al., 1994)

$k'_{2f} = \beta - \beta_e^{(-\delta[H_2O])}$

$\beta = 1.15 \times 10^6 \ (s^{-1})$

$\delta = 0.13 \ (M^{-1})$

$\frac{k_3}{k_{2b}} = 0.06$

$\frac{k_4}{k_{2b}} = 29$

*We have clarified it clearly as follows.*

*"Our findings suggest the need for more direct field quantifications of $\gamma(N_2O_5)$ and the laboratory measurements under extreme ALWC conditions to re-evaluate the response coefficients between $\gamma(N_2O_5)$ and aerosol chemical compositions in parameterizations."*

*And the explanation has been added in section 3.4 as follows.*

*"This phenomenon was possibly caused by several aspects, including the inaccurate estimation of response coefficients of aerosol compositions (represents the quantitative relationship between $\gamma(N_2O_5)$ and aerosol chemical compositions), relative rates of competitive reactions, and the missing parameters."*

Line 91-95: "However, the comparisons of ...." Please rephrase this sentence, as it reads unclearly and awkwardly.

*It has been revised as follows.*

"However, these parameterizations usually exhibit low correlations with observed $\gamma(N_2O_5)$ in varying environments (Brown et al., 2009; Ryder et al., 2014; McDuffie et al., 2018)."

Line 96: "can leads to" should be "can lead to".

Line 105: "European" should be "Europe".

*The above two comments have been revised accordingly in the main text.*

Line 113-114: "We further notice …" the whole sentence reads awkwardly. Please rephrase it.

*It has been revised as follows.*

"We further observe significant biases when estimating particulate nitrate formation potential based on current $\gamma(N_2O_5)$ parameterization."

Line 126: What does CNST mean?

*We have clarified it clearly as follows.*

"Sunrise was around 06:30 CNST (Chinese National Standard Time = UTC + 8 h) and sunset was at 19:30 CNST."

Line 129: "Google Maps images" should it be "Google Map images" ?

Line 138: "and" should be "which".

Line 147: "A total of 117 kinds of VOCs …" could be like "A total of 117 VOCs species …".

Line 155: "included" should be "including".

Line 158: For the header of Table 1, "Detection of limit" should be "Detection limits", "Method" should be "Methods", "Accuracy" should be "Accuracies".

*The above five comments have been revised accordingly in the main text.*

Line 160: "The" is not needed in front of "measurement", for this case. The same applies to the title of the other sections.

*We have revised titles to* "2.3 Measurement and calculation of $\gamma(N_2O_5)$, 2.4 Calculation of $NO_3$ and $N_2O_5$ reactivity, 2.5 Calculation of nitrate production rate, 3.4. Impact of ALWC on parameterized $\gamma(N_2O_5)$"

Line 165: "(excess)" should be "(in excess)".

Line 166: "consist of" should be "consists of".

Line 170: "concentration" should be "concentrations".

*The above three comments have been revised accordingly in the main text.*

Line 169-176: These three sentences read awkwardly and apparently have several grammatical mistakes. Please rewrite them.

*We rewrite it as follows.*

"The detection instruments used for measurements of $N_2O_5$, $NO_x$, $O_3$ and $S_a$ are CEAS-PKU, Thermo 42i-TL, Teledyne T265 and SMPS (TSI Model 3938). Additionally, a RH&T sensor (Rotronic, Model HC2A-S) was utilized to monitor relative humidity and temperature inside the flow tube. During each duty cycle, $N_2O_5$ concentrations were recorded both at the inlet and exit of the flow tube under the condition with and without aerosols to derive the wall loss of $N_2O_5$. NO, $NO_2$, and $O_3$ concentrations were consistently measured at the inlet of the flow tube, and Sa concentrations were consistently measured at the exit of the flow tube."

Line 192: "the reaction rate of constant of…" should be "the reaction rate constant of …".

*We have revised it accordingly in the main text.*

Line 209: "The mean diurnal of measured…" should be "The mean diurnal variation of the measured …". The authors seem to have a big problem on how to correctly use "the".

*We sincerely appreciate your careful check. We have revised it accordingly and also check this problem in our manuscript thoroughly.*

Line 216: "(PNO₃)" should be "P(NO₃)".

Line 218: "observation campaign" is not the common way. Either "observation period" or just "campaign".

*The above two comments have been revised accordingly in the main text.*

Line 252: "Functional dependence" reads very awkwardly. Please change it.

*We have revised this title to* "3.2. Dependence of $\gamma(N_2O_5)$ on impacting factors".

Line 254: What is "organic wet mass faction"? How did you measure it?

*We added the explanation in supporting information as follows.*

"S2. Calculate of organic wet mass fraction.

The organic wet mass fraction is defined as the mass fraction of organics within the aerosol containing water. The calculation of organic wet mass fraction is presented as follows.

Organic wet mass fraction=Organic mass/(Organic mass+$NO_3^-$ mass+$Cl^-$ mass+$SO_4^{2-}$ mass+$NH_4^+$ mass+$H_2O$ mass)"

Line 260-262: As the authors stated, "the negative correlation of particulate organic and γ(N2O5) was usually weak derived from field measurements". In the present work, however, the authors found a significant negative correlation between the organic wet mass fraction and the gamma. Could the authors discuss why this study shows such a difference?

*Thanks for your valuable comments. We added the discussion in section 3.2 in the main text as follows.*

"The organic wet mass fraction in this study varies between 0.3 and 0.8, while other previous studies have reported a variation range of 0.1 to 0.5(McDuffie et al., 2018;Wang et al., 2020;Brown et al., 2009). The large proportion and variation range of organics in the aerosols may lead to a more significant inhibition effect on $\gamma(N_2O_5)$. Additionally, we found that both the dry and wet mass fractions of organics in this study

showed significant negative correlations with ALWC, with Pearson coefficients of -0.66 and -0.79 (Table S3), respectively. Therefore, organics might decrease γ(N2O5) by forming an organic coating to limit the penetration of liquid water into the particle phase and hinder the reaction between $N_2O_5$ with liquid phase."

Line 275: "showed" should be "shows".

*This comment has been revised accordingly in the main text.*

Line 290: In the caption of Fig. 3, the authors state that "The points represent the median in each bin,…". How do the authors select the different bins? Should be the symbols or the points represent the median values?

*We basically divided the range of each aerosol chemical component equally into six bins while with some exceptions. The organic dry mass fraction exhibits a discontinuity beyond 0.6 and thus we set the sixth bin at 0.7. Due to the discontinuity in aerosol nitrate concentration changes beyond 3.5 M, an additional bin was added at 4.75 M. Similarly, since mol Cl⁻/mol NO₃⁻ exhibits a discontinuity beyond 0.4, the sixth bin was positioned at 0.6. We revised the statement of Figure 3 as follows.*

"The gray points represent the measured values. The symbols in different colors represent the median in each bin with range from the 10th to 90th percentile in each bin denoted as lines."

Line 338: "(aka. low …)", for me it is the first time to see the use of "aka.". Probably it is not the official way to use it in a scientific research paper?

*This sentence has been revised as follows.*

"Although some parameterizations performed relatively well in reproducing the median values of γ(N₂O₅), none of the ten parameterizations were able to reproduce the range of measured γ(N₂O₅) values, as indicated by poor correlations and large RMSE."

Line 339-340: The authors use "response coefficients" several times throughout the manuscript. At least it is not a scientifically meaningful definition, as far as I know. What

is the exact meaning of this? Any references?

*Thanks for your comment. We have provided a detailed explanation regarding this issue in the response for Lines 41-42, and we sincerely hope it solves your concern. Response coefficient means the coefficients between $\gamma(N_2O_5)$ and aerosol chemical compositions in parameterizations. Such as $k_3$ and $k_{2b}$ in BT09 (Eq R1). It corresponds to the relative rates of competing reactions(Yu et al., 2020), or represents relationships between aerosol chemical components and $\gamma(N_2O_5)$(McDuffie et al., 2018). The explanation has been added in section 3.4 as follows.*

"This phenomenon was possibly caused by several aspects, including the inaccurate estimation of response coefficients of aerosol compositions (represents the quantitative relationship between $\gamma(N_2O_5)$ and aerosol chemical compositions), relative rates of competitive reactions, and the missing parameters."

Line 358-361: Please rephrase the whole sentence to make it more scientifically readable.

*This sentence has been revised as follows.*

"The overestimation at low ALWC and underestimation at high ALWC suggest that the treatment of coefficients related to ALWC in most parameterizations can hardly capture the response of $\gamma(N_2O_5)$ to largely varied ALWC."

Line 377-379: Same as above

*This sentence has been revised as follows.*

"Therefore, we suggest that future studies should conduct more $\gamma(N_2O_5)$ measurements under extreme ALWC levels, which helps to improve the reliability of response coefficients between $\gamma(N_2O_5)$ and ALWC in ambient conditions."

**References**

[revised manuscript text omitted]